# Hematites Precipitated in Alkaline Precursors: Comparison of Structural and Textural Properties for Methane Oxidation

**DOI:** 10.3390/ijms23158163

**Published:** 2022-07-25

**Authors:** Marta Valášková, Pavel Leštinský, Lenka Matějová, Kateřina Klemencová, Michal Ritz, Christian Schimpf, Mykhailo Motylenko, David Rafaja, Jakub Bělík

**Affiliations:** 1Institute of Environmental Technology, CEET, VSB-Technical University of Ostrava, 17. listopadu 15/2172, 708 00 Ostrava, Czech Republic; pavel.lestinsky@vsb.cz (P.L.); lenka.matejova@vsb.cz (L.M.); katerina.klemencova@vsb.cz (K.K.); 2Department of Chemistry and Physico-Chemical Processes, Faculty of Material Science and Technology, VSB-Technical University of Ostrava, 17. listopadu 15/2172, 708 00 Ostrava, Czech Republic; michal.ritz@vsb.cz; 3Institute of Materials Science, Technical University Bergakademie Freiberg, Gustav Zeuner Street 5, D-09599 Freiberg, Germany; schimpf@iww.tu-freiberg.de (C.S.); motylenko@iww.tu-freiberg.de (M.M.); rafaja@iww.tu-freiberg.de (D.R.); 4RPG Recycling, s.r.o., Member of REC Group, Vazová 2143, 688 01 Uhersky Brod, Czech Republic; belik@rpgrecycling.cz

**Keywords:** hematite nanoparticles, alkaline precipitators, hydrohematites, oxygen carrier, methane oxidation

## Abstract

Hematite (α-Fe_2_O_3_) catalysts prepared using the precipitation methods was found to be highly effective, and therefore, it was studied with methane (CH_4_), showing an excellent stable performance below 500 °C. This study investigates hematite nanoparticles (NPs) obtained by precipitation in water from the precursor of ferric chloride hexahydrate using precipitating agents NaOH or NH_4_OH at maintained pH 11 and calcined up to 500 °C for the catalytic oxidation of low concentrations of CH_4_ (5% by volume in air) at 500 °C to compare their structural state in a CH_4_ reducing environment. The conversion (%) of CH_4_ values decreasing with time was discussed according to the course of different transformation of goethite and hydrohematites NPs precursors to magnetite and the structural state of the calcined hydrohematites. The phase composition, the size and morphology of nanocrystallites, thermal transformation of precipitates and the specific surface area of the NPs were characterized in detail by X-ray powder diffraction, transmission electron microscopy, infrared spectroscopy, thermal TG/DTA analysis and nitrogen physisorption measurements. The results support the finding that after goethite dehydration, transformation to hydrohematite due to structurally incorporated water and vacancies is different from hydrohematite α-Fe_2_O_3_. The surface area *S*BET of Fe_2_O_3__NH-70 precipitate composed of protohematite was larger by about 53 m^2^/g in comparison with Fe_2_O_3__Na-70 precipitate composed of goethite. The oxidation of methane was positively influenced by the hydrohematites of the smaller particle size and the largest lattice volume containing structurally incorporated water and vacancies.

## 1. Introduction

Nanosized iron oxides and oxyhydroxides have been widely studied due to their unique physical properties and wide range of potential applications including magnetism and photocatalytic reactions [1,2], lithium-ion battery and gas sensors [3,4,5] and in catalytic water splitting [6,7,8,9]. Hematite (α-Fe_2_O_3_) is the most thermodynamically stable form of iron oxides and is a potentially interesting catalyst for the complete oxidation of methane with excellent stable performance below 500 °C in both nano and bulk forms. Hematite-like catalysts prepared using the precipitation methods was found to be highly effective, and therefore, it was studied with methane (CH_4_) combustion [10,11,12].

The hematite nanoparticles are obtained either on the wet or dry synthesis. Based on the valence state of Fe, the synthesis includes oxidation from Fe(II) to Fe(III) with oxidants and precipitants [13,14] or the direct preparation of α-Fe_2_O_3_ from Fe(III) precursors under various experimental conditions [15,16,17]. The performance of α-Fe_2_O_3_ strongly depends on the particle size, morphology and structure which are affected by many factors, such as the reactant concentration, the solution pH, the reaction time and temperature and the nature of iron salts [18,19]. The synthesis methods are based on various precursors of iron salts (chlorides, nitrates, sulfates, etc.) as well as different precipitating agents (such as ammonium carbonate, ammonia, sodium hydroxide, urea, etc.), and among them, the precipitation procedure involving hydrothermal synthesis in ferric chloride solutions is the most preferable method [20,21]. The aquatic systems containing FeCl_3_ and a strong base NaOH or a weak base NH_4_OH are favorable for the formation of oxohydroxide phases such as goethite (α-FeOOH), akaganeite (β-FeOOHCl_0.125+x_) and ferrihydrite (Fe_10_O_14_(OH)_2_ [22,23]. The presence of chloride ions in the akaganeite structure with the chemical formula of β-FeO_0.833_(OH)_1.167_Cl_0.167_ [24] is considered as a leftover of high concentrations of chloride in an acidic environment at the early stages of precipitation due to the relatively strong binding to the iron oxyhydroxide precursors.

The synthetic Fe(III) oxides of the composition Fe_2_O_3_ prepared by precipitation and hydrothermal procedures are hematite-like materials not related to pure stoichiometric hematite since the reactions are taking place at the transition sequence from goethite to protohematite to hydrohematite to stoichiometric hematite [23,25,26,27]. The transformation of hydrohematite into stoichiometric hematite includes loss of hydroxyls (OH groups) and residual vacancies, which is accompanied by a decrease in the *c*-axis during the expansion of the *a*-axis [27].

The direct transformation of goethite when assuming an immediate transition of Fe(III) without intermediate stages proceeds according to the chemical Equation (1): 2α-FeOOH → α-Fe_2_O_3_ + H_2_O(1)

When hematite undergoes further thermal treatment, which was stoichiometric from the beginning, only intracrystalline evolutions occur [26]. The structural OH groups and resulting vacant sites in the deformed crystal structure of protohematite-hydrohematite-hematite are given by the general stoichiometric Formula (2) [27], which may play role in the dissociation of water and in the formation of hydroxyl base sites on the surface of nanocrystals:Fe_(2−x/3)_O_(3−x)_(OH)_x_ ∙ *n*H_2_O,(2)

Hematite was investigated as the oxygen carrier for the fuel conversion with easy CO_2_ separation that is almost not influenced by reactivity. Catalytic oxidation of methane has become one of the most effective ways to reduce the low methane concentration by choosing a suitable catalyst that will reduce the activation energy of the reaction and make it a flameless reaction at a lower ignition temperature [28]. The oxidation of methane can be proceeded by the adsorption of oxygen on the oxygen carrier of Fe_2_O_3_ and the catalytic oxidation of methane after complete oxidation [29]. During the catalytic oxidation of methane, some carbon was deposited on the catalysts. On the contrary, less active for combustion than hematite was the transformation of hematite to magnetite (containing Fe(II) and Fe(III)) [11].

The combustion of methane is somewhat complicated because it is necessary to initiate oxidation at a quite high temperature. Therefore, there is a need to dispose of unburned CH_4_ under stringent combustion at low temperatures (<500–550 °C), as described in Equation (3) [12,30]:CH_4_(g) + 2O_2_(g) → CO_2_(g) + 2H_2_O(g)(3)

The chemical reaction of Fe_2_O_3_ with CH_4_ takes place in the decomposition of CH_4_ to carbon and hydrogen (Equation (4)) and the reduction of iron oxide by hydrogen. The possible reaction mechanism was summarized (Equation (5)) by [31]: 3CH_4_ → 3C + 6H_2_(4)
CH_4_ + 12Fe_2_O_3_ → 8Fe_3_O_4_ + CO_2_ + 2H_2_O(5)

This work compares the structural and textural properties of hematite nanoparticles (NPs) prepared using precipitation procedure in water from Fe(III)-chloride hexahydrate precursor and NaOH and NH_4_OH precipitating agents at pH 11. In the experimental process, the different transformation mechanisms of precipitates after calcination at temperatures of 250, 400 and 500 °C that affect the phase composition, surface state and reactivity of the NPs was verified. The aim was to provide evidence that precipitated and calcined hematites are hydrohematites at different structural oxo-hydroxo stage and evaluate them in the function of the oxygen carrier for the oxidation of low concentrations of methane (5% by volume) in air at 500 °C. 

## 2. Results and Discussion

### 2.1. X-ray Diffraction Analysis of NPs

Precipitation of Fe(III) oxide-hydroxides in alkaline aqueous environments from Fe(III) chloride proceeds in equilibrium with ferrihydrite (5Fe_2_O_3_∙9H_2_O) strongly dependent on pH [19,22,32]. Ferrihydride (Fh) was characterized as an important reactive metastable form of Fe(III) oxide-hydroxide phase in aqueous environments which precipitated and turns rapidly into goethite and hematite. The crystalline phases on the XRD patterns in Figure 1 are compared for the precipitates Fe_2_O_3__Na-70 and Fe_2_O_3__NH-70 and after calcination temperatures. 

#### 2.1.1. XRD Phase Analysis of Precipitates 

Fe_2_O_3__Na-70 precipitate obtained using the high alkaline FeCl_3_-NaOH system) was composed of goethite, α-FeOOH (PDF No. 01-073-6522) (Figure 1a). Hematite can be identified only according to the most intensive peak (104). The phase composition agrees with the previous finding about the maximum formation of goethite and hematite at maximum dissolved monovalent Fe(III) ions of ferrihydrite in hydroxo ions solution at a high alkali pH of 12 [32].

Fe_2_O_3__NH-70 precipitate obtained using the FeCl_3_-HN_4_OH system was composed of hematite (PDF No. 01-073-8432), goethite and akageneite, β-FeO(OH) (PDF No. 00-034-1266) (Figure 1a). The composition is in agreement with the previous experimental results documented on minimum concentrations of monovalent Fe(III) ions at pH 8.2, which supported maximum formation of hematite [32]. Akaganeite has also been characterized as the first phase formed at 100 °C, which changed to an OH-rich and Fe-deficient hydrohematite in situ [33].

#### 2.1.2. XRD Phase Analysis of Calcined Precipitates 

Fe_2_O_3__Na-250 sample still contained goethite, ferrihydrite, Fh (PDF No. 00-058-0898) and hematite (Figure 1b). 

The experimental results correspond to the findings published so far. During the thermal dehydration of goethite, Wolska and Schwertmann [26] inferred that protohematite initially forms at about 250 °C. A non-uniform broadening of the XRD peaks of goethite was attributed to the appearance of disordered iron vacancies with a concomitant substitution of OH^−^. Topotactic transformation to hematite can be expected after dehydration and the rearrangement of the solid ferrihydride [32].

Fe_2_O_3__NH-250 sample was composed of hydrohematite (PDF No. 01-073-8433) and bits of goethite were found (Figure 1b). 

Experimental studies suggested that the thermal dehydration process induced topotactic transformation of α-FeOOH either directly to α-Fe_2_O_3_ (1) and/or via two transitional stages of hematite-like intermediate phases identified during dehydroxylation. Topotactic transformation was documented on the hexagonally close packed arrays of anions (O^2−^ or OH^−^) in goethite and preserved in hematite, e.g., [34,35]. Zhang et al. [35] documented that the transformation started preferentially with dehydration on the surface associated with the formation of empty spaces. The transformation process involves hydrogen migration with the formation of adsorbed water, followed by desorption of the water molecules. The calculated values of barriers to hydrogen migration and water desorption indicated a direct reaction pathway without the formation of intermediates.

Precipitates calcined at 400 °C gave Fe_2_O_3__Na-400 and Fe_2_O_3__NH-400 samples containing hydrohematite (PDF No. 01-076-0182 (Fe_1.83_(OH)_0.50_ O_2.50_) (Figure 1c). Similarly, precipitates calcined at 500 °C gave Fe_2_O_3__Na-500 and Fe_2_O_3__NH-500 samples containing hydrohematite and bits of magnetite (Fe_3_O_4_, PDF No. 01-080-7683) (Figure 1d). 

The determination of lattice constants of hematites (Figure 2) confirms the variations in *a* and *c* parameters after calcination at 400 and 500 °C. The Fe_2_O_3__NH-70 precipitate contained hematite with the nonstoichiometric large lattice parameters of *a*, *b* = 5.0345 ± 0.0002 Å and *c* = 13.7646 ± 0.0005 Å, corresponding to the protohematite [26]. 

It is generally accepted that when hematite is precipitated from aqueous solution, protohematite forms initially and exhibits the largest lattice volume due to structurally incorporated water [27]. The subsequent thermal transition of protohematite and goethite and akagenite to hydrohematite involves the loss of molecular water, which will result in a reduction of the unit-cell parameter *c* [26], as can be observed on the *c* = 13.756 ± 0.0002 Å in Fe_2_O_3__NH-500 (Figure 2). 

Goethite in Fe_2_O_3__Na-70 precipitate was transformed in Fe_2_O_3__Na-250 to ferrihydrite and a bit of hematite. Fe_2_O_3__Na-400 and Fe_2_O_3__Na-500 are hydrohematites. Their lattice parameters are smaller in comparison with these parameters of hydrohematites in Fe_2_O_3__NH-400 and Fe_2_O_3__NH-500 (Figure 2). 

### 2.2. Size and Morphology of the Calcined NPs

In sample Fe_2_O_3__Na-500, the bright-field TEM (Figure 3a) disclosed facetted grains with a size of about 50 nm and clusters of very small, pulverulent particles. According to the SAED#1 (Figure 3b), the large grains are single crystals of hydrohematite. This result is supported by the Williamson–Hall analysis of the XRD line broadening [36] which revealed for hydrohematite a mean crystallite size of (49.5 ± 0.8) nm and a very low microstrain (variation of the interplanar spacings, $e=\sqrt{{\left(\Delta d/d\right)}^{2}}$) = (6 ± 1) × 10^−4^. The SAED pattern (Figure 3c) taken on a cluster of small pulverulent particles (SAED#2 in Figure 3a) indicated the presence of a mixture of hydrohematite and ferrihydrite.

Sample Fe_2_O_3__NH-500 contained hydrohematite as a single phase. The hydrohematite particles were slightly smaller (Figure 4a) than the large particles in sample Fe_2_O_3__Na-500. The analysis of the XRD line broadening revealed a mean crystallite size of (37.6 ± 0.3) nm and a microstrain of (9 ± 1) × 10^−4^. As the crystallite size determined from the XRD line broadening agrees very well with the particle size obtained from the TEM micrographs, it can be concluded that also in this sample, the grains are single crystals (cf. Figure 4b,c).

In both precipitation systems, elongated particles were observed, which probably form via oriented growth. Possible mechanisms of the oriented growth of nanocrystals were discussed recently in Ref. [37]. The orientation analysis revealed that the hydrohematite nanoparticles grow preferentially along the basal planes (001). This growth direction can be seen directly in Figure 4c and Figure 5, and is validated by the SAED pattern Figure 4b, because in the crystal structure of hydrohematite, the lattice planes ($2\bar{1}\bar{6}$) and (010) are mutually perpendicular. 

### 2.3. Thermal Transformation of Precipitates

The thermal transformation of the Fe_2_O_3__Na-70 and Fe_2_O_3__NH-70 precipitates was compared by DTA/TG curves (Figure 6). The temperature between 150 and 200 °C is sufficient for rapid conversion of FeOOH polymorphs and may, therefore, involve goethite and akageneite dehydroxylation [33]. 

The thermal dehydration of goethite yields products with hematite structure, which contain similar amounts of water of an intracrystalline character [26]. The endotherm at about 310 °C originates from the release of physically adsorbed water on the surface of goethite. The total weight loss from 50 to 1000 °C was about 10 wt.% in Fe_2_O_3__NH-70 and 10.8 wt.% in Fe_2_O_3__Na-70, and is comparable with the literature (e.g., [38]). 

Crystallization of hydrohematites on the DTA curves of Fe_2_O_3__NH-70 and Fe_2_O_3__Na-70 took place at exothermic maxima around 461 °C and 583 °C, respectively. The crystallization temperatures lower by 122 °C in Fe_2_O_3__NH-70, confirming the presence of hydrohematites in Fe_2_O_3__NH-250, which are identified by the XRD (Figure 1b). 

### 2.4. Infrared Spectroscopy Analysis of NPs

Infrared spectra of NPs samples contain two significant regions: the region of the stretching vibration of hydroxyl (above 3000 cm^−1^), which can provide important information about changes of water during thermal treatment (Figure 7), and the region of the deformation hydroxyl vibration and lattice vibration of Fe–O (below 1000 cm^−1^), which is sensitive to the changes in lattice and hydroxyl bonds (Figure 8). The overlapping bands at both abovementioned regions were refined using the spectral deconvolution procedure. 

(a)The stretching hydroxyl vibration region

The band at 3470 cm^−1^ (observed only in the spectra of samples Fe_2_O_3__Na-70 and Fe_2_O_3__NH-70) was attributed to the H–O–H stretching vibration of non-stoichiometric hydroxyl units of the excess water in iron oxides/oxyhydroxides [39,40]. 

The broad band at 3200–3100 cm^−1^ (observed at all samples) was attributed to the stretching hydroxyl bands of the vibration of O–H group associated with the oxygen atoms of the Fe–O bond in the iron oxides/oxyhydroxides or hydrohematite [39]. The intensity of this band in samples of both precipitation systems with the increasing temperature decreased similarly. An intensive decrease in intensity at 250 °C in the samples Fe_2_O_3__Na-250 and Fe_2_O_3__NH-250 was due to the crystallization of hydrohematite. The third very broad spectral band at 3400–3300 cm^−1^ was assigned to the O–H stretching vibration band of water adsorbed in potassium bromide from the KBr pressed disk [40].

(b)The deformation hydroxyl vibration and lattice vibration in the Fe–O region 

The spectral bands of the deformation vibration of O–H are in the region at 900–800 cm^−1^, and the bands of deformation vibration of Fe–O lattice in the region below 800 cm^−1^ (Figure 8).

The bands of deformation vibration of O–H at 900–800 cm^−1^ were generally used to specify the polymorph of the iron oxides/oxyhydroxides [39,40,41,42,43]. The band at 900 cm^−1^ was assigned to the vibration of out of the mirror plane of goethite (α-FeOOH) and the band at 800 cm^−1^ to the vibration in the mirror plane of goethite [43]. 

The band at 830 cm^−1^ in the spectrum of the sample Fe_2_O_3__NH-70 was assigned to akaganeite (*β*-FeOOH) [42]. The bands of goethite were identified in the spectra of the samples Fe_2_O_3__Na-70, Fe_2_O_3__Na-250 and Fe_2_O_3__NH-250 and were not observed in the samples calcined at temperatures higher than 400 °C. 

The region of the Fe–O lattice vibration at about 480–420 cm^−1^ was assigned to the lattice vibration of FeO_6_ octahedra [43]. The vibration of iron oxides/oxyhydroxides was shifted into lower wavenumbers close to 400 cm^−1^ [42], while the vibration of iron oxides was shifted to the higher wavenumbers (480–450 cm^−1^) [44]. The part of the spectra at about 580–500 cm^−1^ was attributed to the Fe–O vibrational transitions in the hematite hexagonal close packed structure [41]. The part of the spectra at about 590–650 cm^−1^ was attributed to the lattice vibration of the FeO_6_ octahedra [43]. The bands near 630 cm^−1^ were assigned to the structural defects of OH group in protohematite or hydrohematite, e.g., replacement by oxygen atoms in lattice of hematite or changes of Fe-O and Fe-Fe distances in lattice of protohematite [44]. The band at about 690 cm^−1^ is characteristic of poorly crystalline iron hydroxides and defective hematite formed from goethite at low temperatures [43,45]. 

IR spectroscopy made it possible to confirm the existence of hydrohematite based on the of view of the Fe–O lattice vibration region below 1000 cm^−1^, especially the band at approx. 550 cm^−1^ assigned to the Fe–O transitions of hematite in the hexagonal close packed structure. The splitting of the two bands at about 540 cm^−1^ and 580 cm^−1^ observed at all samples indicated the presence of hydrohematite. The higher amount of hydrohematite according to the region above 3000 cm^−1^ in the samples prepared using the precipitation system FeCl_3_-NH_4_OH can be assumed.

### 2.5. Textural Properties (SBET, Vnet) of NPs

The changes within the character of porous structure of hydrohematite nanoparticles are performed in Figure 9 from the nitrogen adsorption-desorption isotherms (Figure 9a,c) as well as pore size distributions (Figure 9b,d). The materials thermally heated at temperatures 70–500 °C show the nitrogen adsorption-desorption isotherms corresponding to the IV type of isotherm with the hysteresis loop getting narrower at higher *p*/*p*_0_ with increasing calcination temperature, which is typical for the mesoporous materials. This general feature related to the effect of calcination temperature on enlarging mesopores size is visible from the evaluated pore size distributions (Figure 9b,d). 

The maxima of the pore size distributions shift from about 3 nm determined for both dried precipitates at 70 °C to about 14.4 nm and 29.3 nm for Fe_2_O_3__Na-400 and Fe_2_O_3__Na-500, respectively (Figure 9b), and to the very similar pore size maxima 21.7 and 22.4 nm for Fe_2_O_3__NH-400 and Fe_2_O_3__NH-500, respectively (Figure 9d).

The influence of the precipitation systems NaOH and NH_4_OH and calcination temperatures on the specific surface areas and pore sizes is performed on the relations in Figure 10. Figure 10a reveals that procedure using NaOH precipitator produced hydrohematite nanoparticles, showing generally higher specific surface areas at all the calcination temperatures used. Since Fe(III)-based material prepared by NH_4_OH at 70 °C shows significantly higher specific surface area of 252 m^2^/g in comparison with NaOH at 70 °C of 199 m^2^/g, more significant changes in the Fe_2_O_3_ nanoparticles morphology prepared by NH_4_OH precipitation procedure can be expected at higher calcination temperature. Similarly, the calcination temperature of 400 and 500 °C produced negligible changes of the pore size when the NH_4_OH precipitator was used. From the point of the larger specific surface area, the nanoparticles prepared using NaOH precipitator are more preferred than the nanoparticles prepared using NH_4_OH precipitator. 

### 2.6. Methane Catalytic Oxidation Test

The reactivity of oxygen in the NPs precipitates Fe_2_O_3__Na-70, Fe_2_O_3__NH-70 and NPs Fe_2_O_3__Na-500, Fe_2_O_3__NH-500 calcined at 500 °C was studied by obtaining light-off curves for the combustion of 5% volume methane in air at a temperature of 500 °C (Figure 11). 

Preliminary blank experiment showed that no methane oxidation occurred in the absence of the catalysts. The absence of carbon monoxide and C_2_ compounds (such as ethane) is consistent with previous works [10,46]. The XRD patterns of catalysts after CH_4_ oxidation for 2 h showed a different presence of hematite and magnetite (Figure 12a). 

After the catalytic test, goethite in Fe_2_O_3__Na-70 and protohematite in Fe_2_O_3__NH-70 were reduced by CH_4_ in Fe_2_O_3__Na-70_cat_ and Fe_2_O_3__NH-70_cat_ (Figure 12a) to the magnetite (PDF No. 01-080-7683) according to (Equation (5)). After 2 h of catalytic experiment, the highest CH_4_ conversion 45.7% to CO_2_ and H_2_O took place on the *S*BET of Fe_2_O_3__NH-70 larger about 53 m^2^/g in comparison with Fe_2_O_3__Na-70 (Table 1). 

The calcined Fe_2_O_3__Na-500 and Fe_2_O_3__NH-500 samples preserved hydrohematite and magnetite in Fe_2_O_3__Na-500_cat_ and Fe_2_O_3__NH-500_cat_ (Figure 12a). The CH_4_ conversion 30.7 and 36.4% was related to the *S*BET 36 and 38 m^2^/g, respectively, and particles size (Table 1). The small amount of hydrogen in the gaseous products and the formation of carbonaceous material on the catalyst surface indicate the catalytic cracking of methane (Equation (4)) [31]. 

In the literature [47], small particles size supported a very fast surface reaction with methane which is followed by slower reactions that progress by oxygen transport from the bulk of the lattice to the surface. In this work, NPs of hydrohematites in Fe_2_O_3__NH-500 are smaller about 20 nm than in Fe_2_O_3__Na-500 and can promote slightly higher CH_4_ conversion (Table 1). 

**Table 1 ijms-23-08163-t001:** Surface area *S*_BET_, net pore volume *V*_net_, pore size *d*p and the mean crystallite size D of samples calcined at 500 °C; Catalytic methane conversion (CH_4_*conv*) after 2 h to CO_2_ and carbon.

Sample	*S*BET (m^2^/g)	*V*net (cm^3^/g)	*d*p (cm^3^/g)	^1^ D (nm)	^2^ D (nm)	CH_4_*conv* (%)	H_2_ (vol.%)	CO_2_ (vol.%)	C_tot_ (wt.%)	C_org_ (%)	C_g_ (%)
Fe_2_O_3__Na-70	199	3	3			39.4	0.009	2.8	0.63	87.9	12.1
Fe_2_O_3__Na-500	36	197	29.3	51.8 ± 15.2	49.5 ± 0.8	30.7	0.021	3.1	0.46	83.6	16.4
Fe_2_O_3__NH-70	252	3	3			45.7	0.004	2.5	0.75	97.9	2.1
Fe_2_O_3__NH-500	38	224	22.4	31.8 ± 4.2	37.6 ± 0.3	36.4	0.006	2.4	1.4	95.8	4.2

^1^ The mean crystallites size D from the XRD line broadening of the diffractions (012) and (104) calculated by Scherrer equation [48] and from ^2^ TEM analysis calculated by Williamson–Hall analysis of the XRD line broadening [36]. C_tot_ = total carbon on catalyst composed of C_org_ = carbon organic and C_g_ = graphitic.

The percentages of methane conversion over hydrohematites at 500 °C (Table 1) are comparable to data in the literature, e.g., the conversion of methane in an amount of 1 vol.% in air on Fe_2_O_3_ at 500 °C did not exceeded 20% [49]. The H_2_, CO and CH_4_ are the reducing agents can convert the weakly magnetic FeOOH and Fe_2_O_3_ to a strongly magnetic phase Fe_3_O_4_ with negative effect on the catalytic activity [31,50].

The NPs precursors Fe_2_O_3__Na-70 (containing goethite, α-FeOOH) and Fe_2_O_3__NH-70 (containing hydrohematite (Fe_1.85_(OH)_0.45_O_2.55_)) (Figure 1a)) were transformed throughout the duration of the experiment to magnetite, Fe_2_O_3_·FeO (Fe_3_O_4_) (Figure 12a). The gaseous products analyzed every 15 min indicated a decreasing CH_4_ conversion with time, which was about 10% lower at the presence of goethite in comparison with hydrohematites (Figure 11). The difference can be explained by the fact that magnetite under reducing conditions was formed easily from goethite, which is unstable at elevated temperatures, in comparison with hematite [51].

XRD patterns of the NPs calcined Fe_2_O_3__Na-500 and Fe_2_O_3__NH-500 containing hydrohematites (Figure 1d) and of Fe_2_O_3__Na-500_cat_ and Fe_2_O_3__NH-500_cat_ (Figure 12a) were similar and phases change cannot be assumed. However, the conversion (%) of CH_4_ in the presence of these calcined catalysts was also decreasing with the time (Figure 11). After 2 h of the experiment, the lattice parameters in the Fe_2_O_3__Na-500_cat_ and Fe_2_O_3__NH-500_cat_ were obviously extended in comparison with these parameters in the Fe_2_O_3__Na-500 and Fe_2_O_3__NH-500 (Figure 12b). In the literature, a negative effect on the catalytic activity was ascribed to the migration of oxygen from the inner bulk to the surface [52] and/or deposition of methane reaction intermediates [12].

This assumption is based on a combined experimental and theoretical study of methane oxidation and reaction to intermediates over hematite [12], which brought findings on a course of reactions as follows: methane is adsorbed on the surface of lattice oxygens bounded to the iron center, forming CH_3_–O species, which then transforms through the reaction intermediates formed via a combination of thermal hydrogen-atom transfer and proton-coupled electron transfer processes. CO_2_ and H_2_O are formed and desorbed, leaving oxygen vacancies on the surface, while other neighboring lattice oxygens and O_2_ from the gas phase replenish the vacancies and reconstruct the active center. Based on these findings, the very intensive decrease of conversion (%) of CH_4_ over time with the catalyst Fe_2_O_3__Na-500 and very intensive expansion of the lattice dimensions in Fe_2_O_3__Na-500_cat_ can be explained.

## 3. Materials and Methods

### 3.1. Materials and Samples Preparation

The synthesis of hematite nanoparticles (NPs) was performed using the ferric chloride hexahydrate (FeCl_3_∙6H_2_O) salt precursor and sodium hydroxide (NaOH) or ammonium hydroxide (NH_4_OH, 28% NH_3_ in H_2_O) as precipitation agents (supplied by the company Lach-Ner Co., Neratovice, Czech Republic). The basic source of hematite NPs were two 100 mL batches of 0.05 M Fe(III) solution. The precipitation was performed by adding dropwise 2M NaOH to the one batch and NH_4_OH (28% NH_3_ in H_2_O) to the other batch until pH 11 under magnetic stirring at 70 °C (Heidolph MR Hei-Tec, Heidolph, Heidolph Instruments GmbH & Co., KG, Schwabach, Germany). The resulting gel products were centrifuged at 6000 rpm and water-washed to free Cl^−^, Na^+^ and NH_4_^+^ ions and then dried at 70 °C for 4 h. Dry Fe_2_O_3__Na-70 and Fe_2_O_3__NH-70 NPs precursors were then calcined at 250, 400 and 500 °C for 4 h. In the further text, the samples synthesized with NaOH and NH_4_OH are denoted as Fe_2_O_3__Na-XXX and Fe_2_O_3__NH-XXX, respectively. The suffix XXX means the calcination temperature.

### 3.2. Methane Catalytic Oxidation Experiment

Reactivity of iron oxide samples with methane was performed on combustion 5 vol.% methane in air at 500° for 135 min. The sample (50 mg) crushed and sieved to 40 μm was loaded between wool plugs in a quartz tubular reactor 600 mm length and 4 mm internal diameter. The reactor was placed inside an electric furnace (LAC, Ltd., Židlochovice, Czech Republic) at a temperature ramp of 10 °C/min to reach 500 °C, and then, it was kept by means of an external PID controller (Papouch store s.r.o., Prague, Czech Republic). The flow rate of 25 mL/min of 5 vol.% methane in air was controlled by a mass flow controller (Aalborg Digital Mass Flowmeter, Merck KGaA, Darmstadt, Germany) over the sample (O_2_/CH_4_ molar ratio = 5:1). The gaseous product collected in 1 L Tedlar Bags was then analyzed every 15 min on gas chromatograph (YL 6100). The carbonaceous materials deposited on the surface of the catalysts was analyzed using an RC612 Multiphase Carbon and Water Analyzer (LECO Instruments, St. Joseph, MI, USA).

### 3.3. Methods Characterization

X-ray diffraction (XRD) patterns were measured and the XRD analyses were carried out on a Rigaku SmartLab diffractometer (RIGAKU Corporation, Tokyo, Japan) working in the symmetrical Bragg–Brentano geometry and with the CoK*α* radiation (*λ*_1_ = 0.178892 nm, *λ*_2_ = 0.179278 nm). The acceleration voltage on the sealed tube was 40 kV, the current 40 mA. The diffracted intensities were recorded by a 1D silicon strip detector D/teX Ultra 250 in the 5°–80° 2*θ* range with the speed of 0.5°/min and a step size of approx. 0.01°. The powder samples were placed on a rotated Si sample holder that was rotated with the speed of 15 rpm. The XRD patterns were evaluated using PDXL2 software No. 2.4.2.0 (Rigaku Corporation, Tokyo, Japan) and compared with database PDF-2, 2015 (ICDD, Newton Square, PA, USA). For selected samples, supplementary XRD measurements were performed with a Seifert/FPM Bragg–Brentano diffractometer (Freiberg, Germany) in the 2*θ* range between 12° and 130° to evaluate the microstrain and to verify the size of hematite crystallites. These XRD measurements (in situ XRD) were carried out with the CoK*α* radiation and with a 1D detector as well.

High resolution transmission electron microscope: The nanoparticles were analyzed with an analytical high-resolution transmission electron microscope (HRTEM) JEM 2200 FS from Jeol (Tokyo, Japan), which was equipped by a corrector of the spherical aberration (Cs) and operated at 200 kV acceleration voltage. The HRTEM analyses comprised the visualization of the size and morphology of the NP, and the local phase identification using selected area electron diffraction (SAED) and high-resolution imaging complemented by the fast Fourier transformation (FFT) of the HRTEM micrographs.

Specific surface area and porosity: The specific surface area and porosity were measured using the 3Flex physisorption set-up (Micromeritics Instrument Corporation, Norcross, GA, USA). The specific surface area *S* was quantified according to the classical Brunauer–Emmett–Teller (BET) theory for the *p/p*_0_ = 0.05–0.25. The mesopore-macropore size distribution was evaluated from the adsorption branch of the nitrogen adsorption-desorption isotherm using the Barrett–Joyner–Halenda (BJH) method, assuming the cylindrical pore geometry (characterized by the diameter *d*_p_ of the pores) and using the Broekhoff–de Boer standard isotherm with Faas correction.

Thermal analysis: The thermogravimetric TG/DTA analysis of the precipitators Fe_2_O_3__Na-70 and Fe_2_O_3__NH-70 was obtained using the Thermal Analyzer SDT 650 Instruments (New Castle, DE, USA) in nitrogen atmosphere (flow 0.1 L/min) between 25 and 1000 °C at a heating rate of 10 °C/min.

Infrared spectroscopy: The infrared (IR) spectra were obtained on a Nicolet iS50 FTIR spectrometer (ThermoScientific, Madison, WI, USA), equipped with KBr beamsplitter and DTGS detector for the mid infrared region (4000–400 cm^−1^). The spectral deconvolution of the spectral bands in the selected spectral region was performed by PeakResolve software (an integral part of spectroscopic software Omnic, ThermoScientific, Madison, WI, USA). The number of overlapped bands was predicted by Fourier deconvolution and by the second derivative operation. Gaussian/Lorentzian mixed function was used for the fitting of the separated bands.

## 4. Conclusions

Hematite nanoparticles prepared using precipitation in alkaline pH 11 conditions and calcination up to 500 °C were characterized as hydrohematites. The precursors of nanoparticles precipitated at 70 °C and calcined at 500 °C were tested as oxygen carrier for methane oxidation at 500 °C. Differences in methane conversion at 500 °C were explained based on the structural properties. The alkaline precipitators play a kay role in the formation of goethite or protohematite, hydrohematite crystallite size, surface *S*BET area, and the catalytic activity and stability on methane oxidation. Hydrohematites prepared from Fe(III) solution with NH_4_OH precipitator exhibited better catalytic effect on methane decomposition than when precipitated with NaOH.

## Figures and Tables

**Figure 1 ijms-23-08163-f001:**
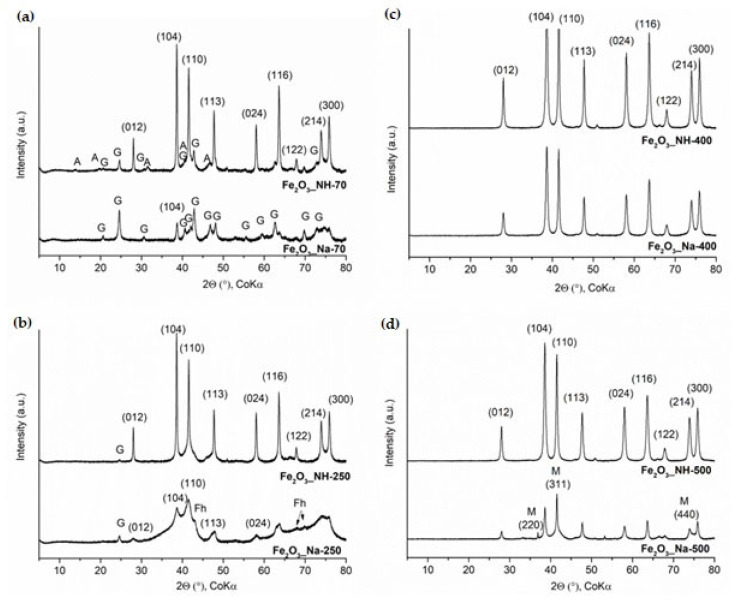
XRD patterns of the hematite samples prepared using two precipitated systems at: (**a**) 70 °C, (**b**) 250 °C, (**c**) 400 °C, (**d**) 500 °C; peaks of hematite are assigned with indices (hkl), G = goethite, A = akageneite, Fh = ferrihydrite, M = magnetite.

**Figure 2 ijms-23-08163-f002:**
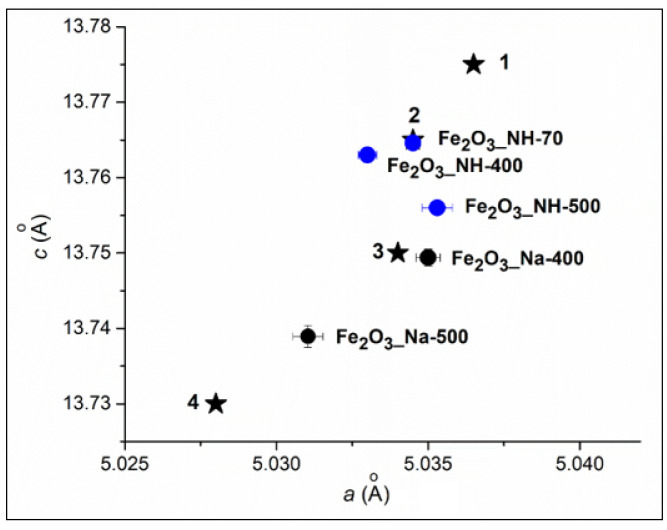
The lattice parameters *c-a* of hydrohematites in nanoparticles prepared using NaOH and NH_4_OH precipitators and calcination at 400 and 500 °C between others parameters of hydrohematies: ★ 1—PDF No. 01-073-8432 (Fe_1.85_(OH)_0.66_O_2.34_), ★ 2—PDF No. 01-073-8433 (Fe_1.85_(OH)_0.45_O_2.55_), ★ 3—PDF No. 01-076-0182 (Fe_1.83_(OH)_0.50_ O_2.50_) and hematite: ★ 4—PDF No. 00-001-1053 (Fe_2_O_3_).

**Figure 3 ijms-23-08163-f003:**
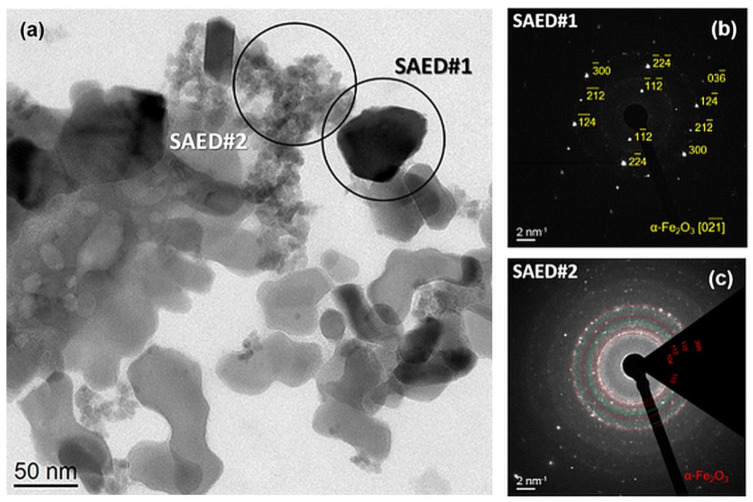
Bright-field TEM image of sample Fe_2_O_3__Na-500: (**a**) SAED pattern of a large hydrohematite particle (**b**) and SAED pattern of the agglomerate of small particle containing a mixture of hydrohematite and ferrihydrite (**c**).

**Figure 4 ijms-23-08163-f004:**
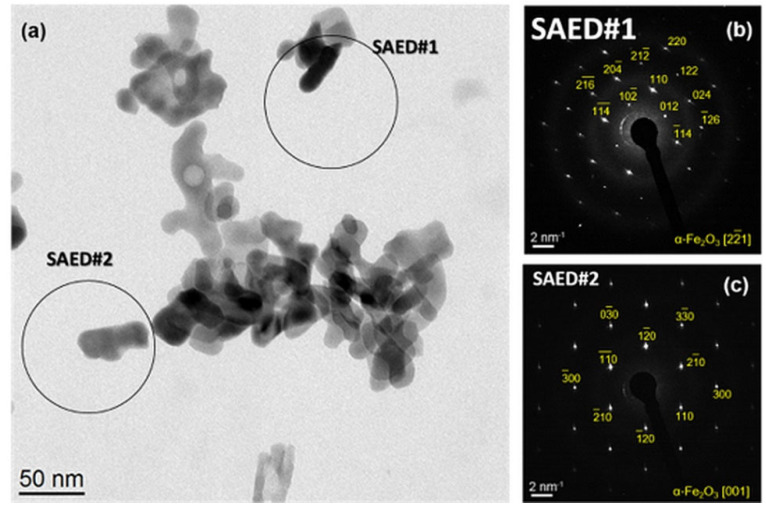
Bright-field TEM image of sample Fe_2_O_3__NH-500 (**a**) and SAED patterns of two differently oriented hydrohematite particles (**b**,**c**).

**Figure 5 ijms-23-08163-f005:**
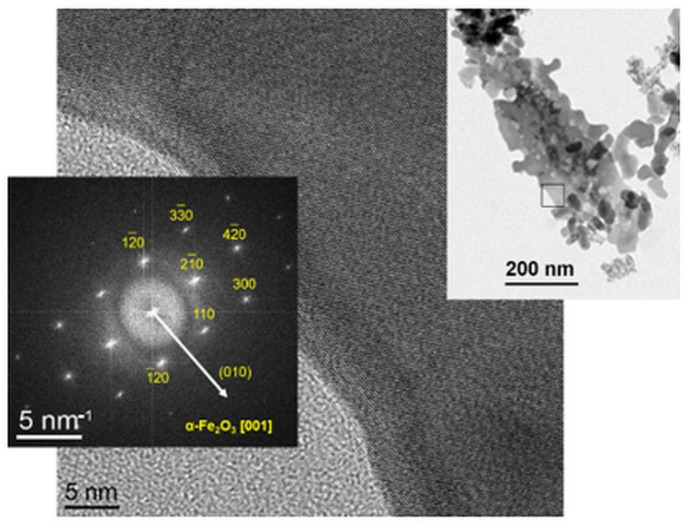
HRTEM micrograph of an elongated hydrohematite particle in sample Fe_2_O_3__Na-500 and its fast Fourier transform. The position of the HRTEM micrograph is marked in the bright-field micrograph displayed in the top right panel. The arrow in the FFT indicates the normal direction to the lattice planes (010).

**Figure 6 ijms-23-08163-f006:**
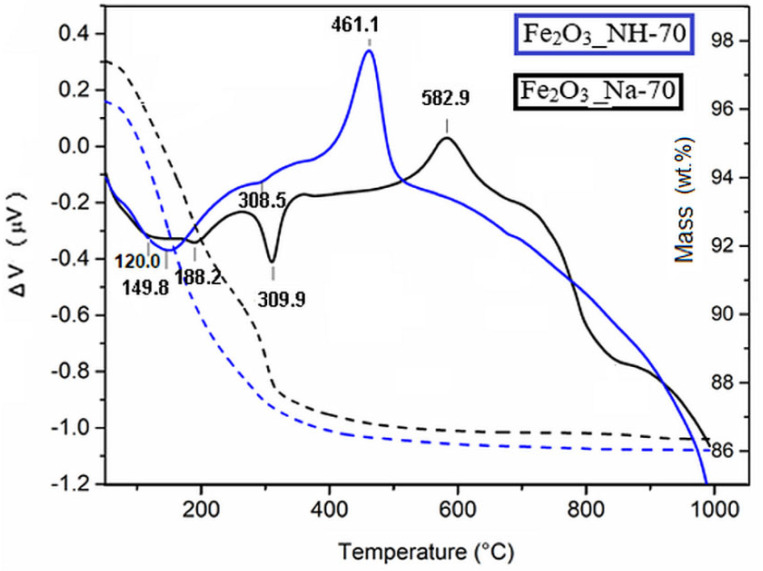
The DTA (full line) and TG (dashed line) curves of the precipitated samples.

**Figure 7 ijms-23-08163-f007:**
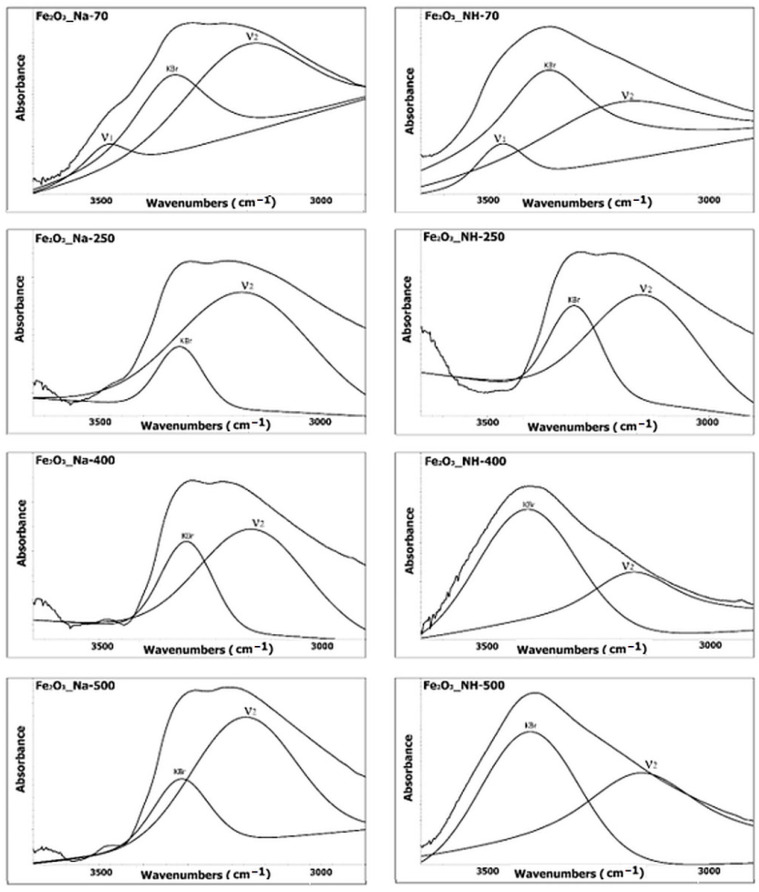
IR spectral bands above 3000 cm^−1^ comprise the stretching vibration regions of water (ν1) and hydroxyls (ν2).

**Figure 8 ijms-23-08163-f008:**
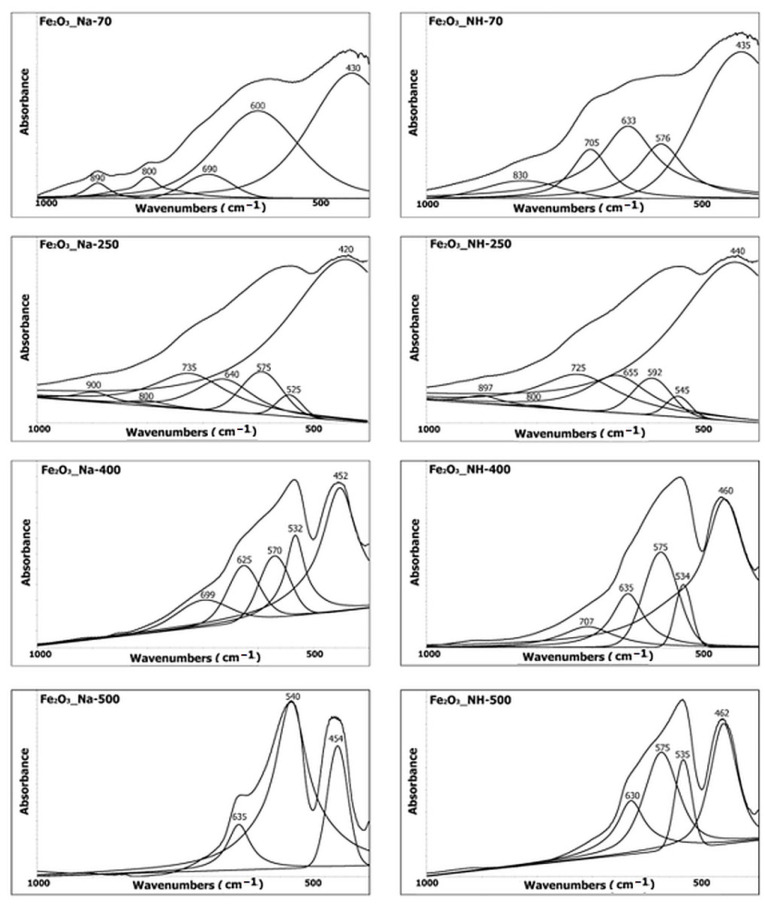
IR spectral bands below 1000 cm^−1^.

**Figure 9 ijms-23-08163-f009:**
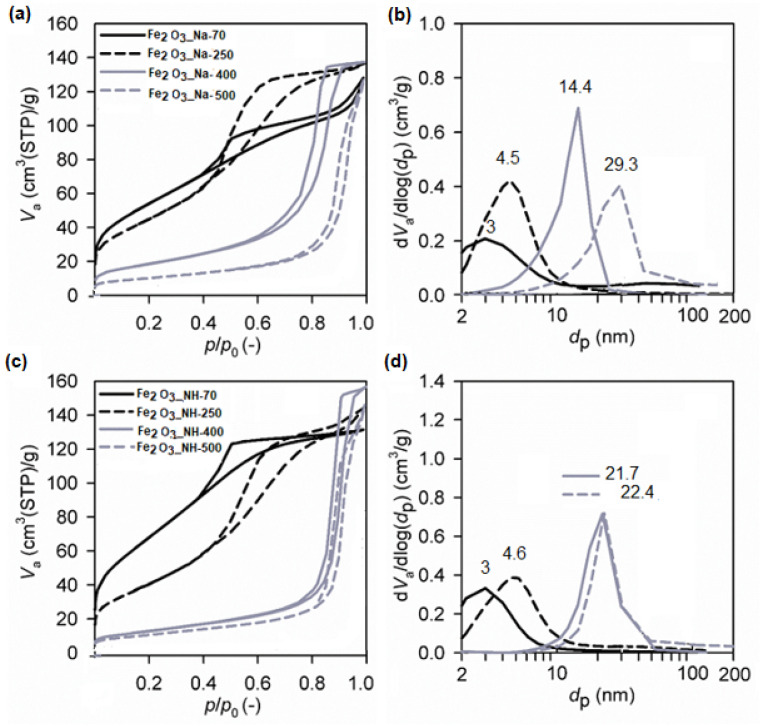
Measured nitrogen adsorption-desorption isotherms and evaluated pore size distributions of Fe_2_O_3_ nanoparticles prepared using the precipitators and calcined at different temperatures: NaOH (**a**,**b**) and NH_4_OH (**c**,**d**).

**Figure 10 ijms-23-08163-f010:**
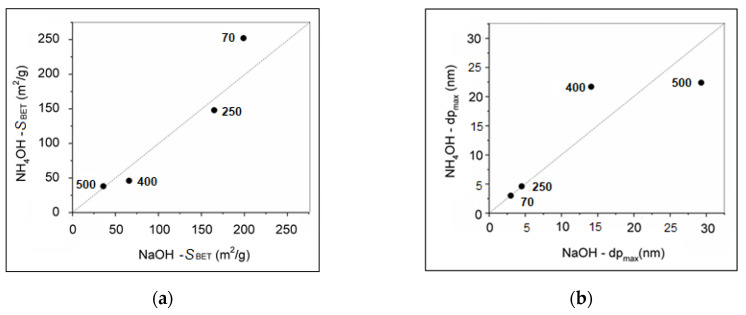
Relation between (**a**) BET specific surface areas and (**b**) pore size maxima of Fe_2_O_3_ nanoparticles prepared using the precipitators NaOH and NH_4_OH and calcined at different temperatures.

**Figure 11 ijms-23-08163-f011:**
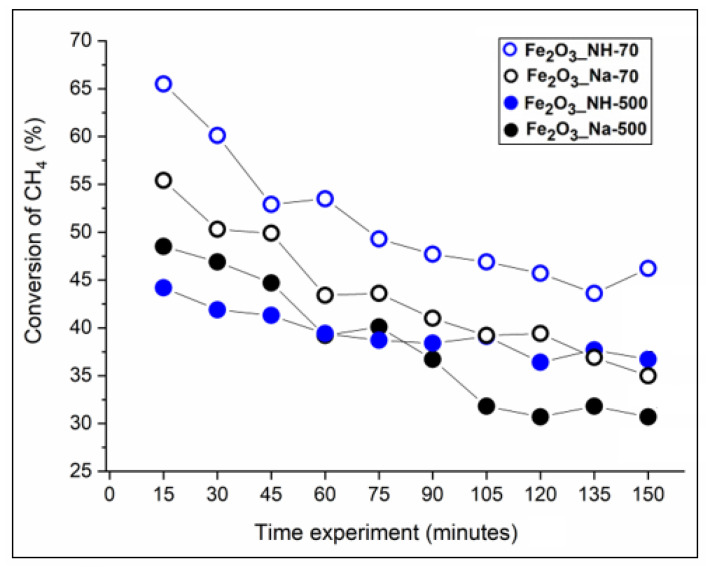
Time conversion of CH_4_ at the presence of NPs precipitated at 70 °C and calcined at 500 °C.

**Figure 12 ijms-23-08163-f012:**
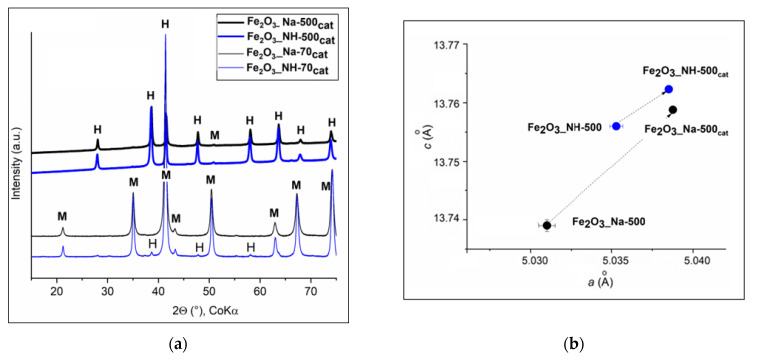
Hydrohematite samples after 2 h of the methane catalytic oxidation test: (**a**) XRD patterns with the peaks denoted as: H = Fe_2_O_3_ and M = Fe_3_O_4_; (**b**) The lattice parameters *a, c* of Fe_2_O_3__Na-500 and Fe_2_O_3__NH-500 catalysts were expanded in Fe_2_O_3__Na-500_cat_ and Fe_2_O_3__NH-500_cat_.

## Data Availability

The data supporting reported results are available on request from the corresponding author.

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
