# Peer review of "Hematites Precipitated in Alkaline Precursors: Comparison of Structural and Textural Properties for Methane Oxidation"

_ijms, 2022, doi:10.3390/ijms23158163_

Round 1
Reviewer 1 Report
The manuscript by Valášková et al., titled "Hematites Precipitated in Alkaline Precursors: Comparison of 2 Structural and Textural Properties for Methane Oxidation" reports on the synthesis of nanohematites obtained with the precipitation method in the presence of NaOH or NH4OH and drying at 70°C, followed by calcination at different temperature under air.
The obtained materials were fully characterized and tested as catalysts for the methane oxidation.
The reported results are interesting, therefore I suggest publication, after major revision.
1) time of calcination should be explained. 3h? 12h? Calcination time is very important. In addition, the preparation procedure should be reported in details, because the cited article talked about synthesis at 80°C and calcination at 700°C. In the present article, it is not clear if the synthesis is carried out at 70°C or 80°C. Moreover, the cited article talked about different starting concentration of FeCl3. Which concentration was used in the present work?
2) Figure 1 should also include XRD patterns of the hematite samples prepared using two precipitated systems and calcined at 500°C for 2 h to be compared to XRD patterns reported in Figure 12a. In addition, the label "hkl" on the peaks is missing.
3) Figure 11 is not clear. It seems that conversion of CH4 is, for example, 65% at time = 0 minute. Please, explain.
4) Figure 12a: please, use four different colors, otherwise it is hard to distinguish between the four catalysts. How can the authors be sure that the diffraction patterns of Figure 11a belong to magnetite instead of maghemite? Infact, Fe3O4 and gamma-Fe2O3 have the same diffraction pattern. Maybe, IR spectra can solve any doubt.
5) Pag. 4: The following sentence : "Precipitates at 400 °C were calcined in Fe2O3_Na-400 and Fe2O3_NH-400 samples to the hydrohematite" does not make any sense. I would convert it into "Materials calcined at 400 °C gave Fe2O3_Na-400 and Fe2O3_NH-400 samples containing hydrohematite" (if I correctly understood). The other sentences should be changed, accordingly.
6) This article reports the synthesis of nanohematite under mild conditions (Figure 1a). A comparison with other methods (Journal of Alloys and Compounds 462 (2008) 24–28; Eur. J. Inorg. Chem. 2022, e202100943) should be cited in the introduction
7) English needs to be polished
Author Response
Reviewer #1: R1 The manuscript by Valášková et al., titled "Hematites Precipitated in Alkaline Precursors: Comparison of 2 Structural and Textural Properties for Methane Oxidation" reports on the synthesis of nanohematites obtained with the precipitation method in the presence of NaOH or NH4OH and drying at 70°C, followed by calcination at different temperature under air.
The obtained materials were fully characterized and tested as catalysts for the methane oxidation.
The reported results are interesting, therefore I suggest publication, after major revision.
Response:
Thank you for your positive comments. We considered the aspects proposed by referee.
Action:
See below for details.
- time of calcination should be explained. 3h? 12h? Calcination time is very important. In addition, the preparation procedure should be reported in detail, because the cited article talked about synthesis at 80°C and calcination at 700°C. In the present article, it is not clear if the synthesis is carried out at 70°C or 80°C. Moreover, the cited article talked about different starting concentration of FeCl3. Which concentration was used in the present work?
Response:
Thank you for pointing out this deficiency. The text was inserted in: 4.1 Materials and Samples Preparation:
Action:
The basic source of hematite NPs were two 100 mL batches of 0.05 M Fe(III) solution. The precipitation was performed by adding dropwise 2M NaOH to the one batch and NH4OH (28% NH3 in H2O) to the other batch until pH 11 under magnetic stirring at 70 °C (Heidolph MR Hei-Tec, Heidolph, Heidolph Instruments GmbH & Co. KG, Germany). The resulting gel products were centrifuged at 6000 rpm and water-washed to free Cl-, Na+, NH4+ ions and then dried at 70 °C for 4 h. Dry Fe2O3_Na-70 and Fe2O3_NH-70 NPs precursors were then calcined at 250, 400 and 500 °C for 4 h. In the further text, the samples synthesized with NaOH and NH4OH are denoted as Fe2O3_Na-XXX and Fe2O3_NH-XXX, respectively. The suffix XXX means the calcination temperature.
- Figure 1 should also include XRD patterns of the hematite samples prepared using two precipitated systems and calcined at 500°C for 2 h to be compared to XRD patterns reported in Figure 12a. In addition, the label "hkl" on the peaks is missing.
Response:
Yes, we added the whole XRD patterns.
Action:
Figure 1. XRD patterns of the hematite samples prepared using two precipitated systems at: (a) 70 °C, (b) 250 °C, (c) 400 °C, (d) 500 °C; peaks of hematite are assigned with indices (hkl), G = goethite, A = akageneite, Fh = ferrihydrite, M = magnetite.
- Figure 11 is not clear. It seems that conversion of CH4 is, for example, 65% at time = 0 minute. Please, explain.
Response:
Thank you for your notice on this figure. We must apologize for the error. Zero time means that at that moment the reactor was heated to a final temperature of 500 °C and the reaction mixture (CH4 + air) was introduced into the reactor using the flow controllers. Until then, only nitrogen was flowing through the reactor. The collection of gaseous products into Tedlar bags was carried out after 15 minutes. All measurements are therefore shifted by 15 minutes (corrected in Table 1).
Action:
Figure 11 has been corrected:
- Figure 12a: please, use four different colors, otherwise it is hard to distinguish between the four catalysts. How can the authors be sure that the diffraction patterns of Figure 11a belong to magnetite instead of maghemite? Infact, Fe3O4 and gamma-Fe2O3 have the same diffraction pattern. Maybe, IR spectra can solve any doubt.
Response: The thickness of the diffraction patterns line of the catalyst after the methane oxidation is highlighted, but with the preservation of the colors due to association with Na and NH4 precipitators.
We are convinced of the correctness of the identification of magnetite by the very precise positions of the diffractions, which are different from maghemite. In any case, Fe2O3.FeO composition in both modifications is here important.
Action:
Figure 12a was corrected.
5) Pag. 4: The following sentence: "Precipitates at 400 °C were calcined in Fe2O3_Na-400 and Fe2O3_NH-400 samples to the hydrohematite" does not make any sense. I would convert it into "Materials calcined at 400 °C gave Fe2O3_Na-400 and Fe2O3_NH-400 samples containing hydrohematite" (if I correctly understood). The other sentences should be changed, accordingly.
Response:
The sentences were changes accordingly.
Action:
Precipitates calcined at 400 °C gave Fe2O3_Na-400 and Fe2O3_NH-400 samples containing hydrohematite (PDF No. 01-076-0182 (Fe1.83(OH)0.50 O2.50) (Figure 1c). Similarly, precipitates calcined at 500 °C gave Fe2O3_Na-500 and Fe2O3_NH-500 samples containing hydrohematite and bits of magnetite (Fe3O4, PDF No. 01-080-7683) (Figure 1d).
- This article reports the synthesis of nanohematite under mild conditions (Figure 1a). A comparison with other methods (Journal of Alloys and Compounds 462 (2008) 24–28; Eur. J. Inorg. Chem. 2022, e202100943) should be cited in the introduction.
Action:
articles are cited under Nos. 16 and 17.

Reviewer 2 Report
The present work reports the structural and textural properties of hematite nanoparticles prepared by precipitation. By studying the products under different reaction conditions, it is proposed that the alkaline precipitators play a key role in the formation of goethite or protohematite, hydrohematite crystallite size, surface SBET area, and the catalytic activity and stability on methane oxidation. Furthermore, it identifies the optimal reaction conditions for catalyst preparation applied to methane combustion. However, for the performance test part of the catalyst applied to methane combustion, there is still a lack of data on stability and catalytic activity. Therefore, a few of points should be checked by the authors in preparing the manuscript for future submissions, as addressed below:
1. It is recommended to place the experimental section before the analysis of the results, so that the readers can understand the meaning of the abbreviations of the experimental sample names. In addition,the alphabet soup is too cumbersome,it is better to use a more concise code, such as “Fe2O3_NH-400”.
2. Page 2, line 82,“a suitable catalyst that will reduce the activation energy of the reaction”. Whether the apparent activation energy of methane combustion on this catalyst is determined? And the catalyst actually lowers the activation energy of the reaction?
3. Page 4, line 152-155, it is mentioned that the Fe2O3_Na-400 and Fe2O3_NH-400 samples are composed of hematite, but there is no XRD pattern to prove it.
4. The maintext states that “During the catalytic oxidation of methane, some carbon was deposited on the catalysts.” However, only the catalysts used after two hours were characterized in this manuscript. Whether a cycle test was performed? How stable is the catalyst?
5. The authors used sodium hydroxide (NaOH) or ammonium hydroxide (NH4OH) as precipitants to study the effect of pH on the particle precipitation process, but it didn’t illustrate the effects of cations (Na+or NH4+).
6. Please notice the normative and aesthetics of the illustrations, as well as typographical issues. For example, it is recommended to unify the size and font of the text in the illustration, etc.
Author Response
Thank you for your positive comments. We are aware that the catalytic properties are not sufficiently implemented in the work. The intention of this work is stated in the title. As a result, the structural properties of these catalysts are compared in detail and, according to them, their different catalytic abilities in methane oxidation are explained. We would like to note that hematite nanoparticles prepared by precipitation procedures are not commonly characterized correctly as hydrohematites with structural hydroxyls and bound water molecules in the lattice and discussed to the explanation of the oxidation ability.
- It is recommended to place the experimental section before the analysis of the results, so that the readers can understand the meaning of the abbreviations of the experimental sample names. In addition,the alphabet soup is too cumbersome, it is better to use a more concise code, such as “Fe2O3_NH-400”.
Response:
Yes, we are of the same opinion, but the template for the IJMS article has been respected here.
Thank you for the notes regarding the labeling of the samples. We take the liberty of not making the change because we kept the same as in our previous published works.
- Page 2, line 82,“a suitable catalyst that will reduce the activation energy of the reaction”. Whether the apparent activation energy of methane combustion on this catalyst is determined? And the catalyst actually lowers the activation energy of the reaction?
Response:
These catalyst properties observed by Ref [26] mentioned in Introduction were in this work not determined.
- Page 4, line 152-155, it is mentioned that the Fe2O3_Na-400 and Fe2O3_NH-400 samples are composed of hematite, but there is no XRD pattern to prove it.
Response:
Figure 1 was corrected.
Action:
Figure 1. XRD patterns of the hematite samples prepared using two precipitated systems at: (a) 70 °C, (b) 250 °C, (c) 400 °C, (d) 500 °C; peaks of hematite are assigned with indices (hkl), G = goethite, A = akageneite, Fh = ferrihydrite, M = magnetite.
- The maintext states that “During the catalytic oxidation of methane, some carbon was deposited on the catalysts.” However, only the catalysts used after two hours were characterized in this manuscript. Whether a cycle test was performed? How stable is the catalyst?
Response:
We are convinced that for the comparison of the effect of structural differences, the test for hydrohematite NPs precursors and calcined at 500 °C is sufficient. A more detailed characterization of these catalysts deserves a special study and publication in a catalytic journal.
- The authors used sodium hydroxide (NaOH) or ammonium hydroxide (NH4OH) as precipitants to study the effect of pH on the particle precipitation process, but it didn’t illustrate the effects of cations (Na+or NH4+).
Response:
Thank you for this comment. The answer is in the additionally inserted text about sample preparation in 4.1 Materials and Samples Preparation:
Action:
The basic source of hematite NPs were two 100 mL batches of 0.05 M Fe(III) solution. The precipitation was performed by adding dropwise 2M NaOH to the one batch and NH4OH (28% NH3 in H2O) to the other batch until pH 11 under magnetic stirring at 70 °C (Heidolph MR Hei-Tec, Heidolph, Heidolph Instruments GmbH & Co. KG, Germany). The resulting gel products were centrifuged at 6000 rpm and water-washed to free Cl-, Na+, NH4+ ions and then dried at 70 °C for 4 h. Dry Fe2O3_Na-70 and Fe2O3_NH-70 NPs precursors were then calcined at 250, 400 and 500 °C for 4 h.
- Please notice the normative and aesthetics of the illustrations, as well as typographical issues. For example, it is recommended to unify the size and font of the text in the illustration, etc.
Response:
Thank you for the notice. Text and typographical issues were unified.

Round 2
Reviewer 1 Report
All suggestions given by reviewers have been followed, thus I suggest publication of the present article, after the following minor revision concerning figure 11.
It is still unclear if the y-axis of Figure 11 reports CH4(%) or conversion (%) of CH4 values. In fact, CH4(%) should decrease with the time, on the contrary conversion (%) of CH4 should increase with the time. Figure 11 showed a decreasing of conversion (%) of CH4 with the time.
If conversion (%) of CH4 really decreased with the time, the authors should better discuss the reasons of such deactivation of the catalysts
Author Response
Author's Reply to the Review Report (Reviewer 1)
Comments and Suggestions for Authors
All suggestions given by reviewers have been followed, thus I suggest publication of the present article, after the following minor revision concerning figure 11.
It is still unclear if the y-axis of Figure 11 reports CH4(%) or conversion (%) of CH4 values. In fact, CH4(%) should decrease with the time, on the contrary conversion (%) of CH4 should increase with the time. Figure 11 showed a decreasing of conversion (%) of CH4 with the time.
If conversion (%) of CH4 really decreased with the time, the authors should better discuss the reasons of such deactivation of the catalysts
Submission Date
30 June 2022
Date of this review
18 Jul 2022 15:44:52
Dear Editor,
We appreciate the comments of referee, and we address our response to the comment concerning figure 11 - revision and discussion.
With best regards,
Marta Valášková
In the abstract the sentence was inserted: The conversion (%) of CH4 values decreasing with time was discussed according to the course of different transformation of goethite and hydrohematites NPs precursors to magnetite and the structural state of the calcined hydrohematites.
We hope the text embedded below is informative enough to explain the relation in Figures 11 and 12.
On the page 16, the text was inserted:
The percentages of methane conversion over hydrohematites at 500 °C (Table 1) are comparable to data in the literature, e.g., the conversion of methane in an amount of 1 vol. % in air on Fe2O3 at 500 °C did not exceeded 20 % [49]. The H2, CO and CH4 are the reducing agents and can convert the weakly magnetic FeOOH and Fe2O3 to a strongly magnetic phase Fe3O4 with negative effect on the catalytic activity [31, 50].
The NPs precursors Fe2O3_Na-70 (containing goethite, α-FeOOH) and Fe2O3_NH-70 (containing hydrohematite (Fe1.85(OH)0.45O2.55)), (Figure 1a)) were transformed throughout the duration of the experiment to magnetite, Fe2O3·FeO (Fe3O4), (Figure 12a). The gaseous products analyzed every 15 minutes indicated a decreasing CH4 conversion with time, which was about 10 % lower at the presence of goethite in comparison with hydrohematites (Figure 11). The difference can be explained by the fact, that magnetite under reducing conditions was formed easily from goethite, which is unstable at elevated temperature, in comparison with hematite [51].
XRD patterns of the NPs calcined Fe2O3_Na-500 and Fe2O3_NH-500 containing hydrohematites (Figure 1d) and of Fe2O3_Na-500cat and Fe2O3_NH-500cat (Figure 12a) were similar and phases change cannot be assumed. However, the conversion (%) of CH4 in the presence of these calcined catalysts was also decreasing with the time (Figure 11). After 2 hours of the experiment, the lattice parameters in the Fe2O3_Na-500cat and Fe2O3_NH-500cat were obviously extended in comparison with these parameters in the Fe2O3_Na-500 and Fe2O3_NH-500 (Figure 12b). In literature, a negative effect on the catalytic activity was ascribed to the migration of oxygen from the inner bulk to the surface [52] and/or deposition of methane reaction intermediates [12].
This assumption is based on a combined experimental and theoretical study of methane oxidation and reaction to intermediates over hematite [12], which brought findings on the course of reactions as follows: Methane is adsorbed on the surface of lattice oxygens bounded to the iron center, forming CH3–O species, which then transforms through the reaction intermediates formed via a combination of thermal hydrogen-atom transfer and proton-coupled electron transfer processes. CO2 and H2O are formed and desorbed, leaving oxygen vacancies on the surface, while other neighboring lattice oxygens and O2 from the gas phase replenish the vacancies and reconstruct the active center. Based on these findings, the very intensive decrease of conversion (%) of CH4 with the time over the catalyst Fe2O3_Na-500 and very intensive expansion of the lattice dimensions in Fe2O3_Na-500cat, can be explained.

Reviewer 2 Report
It could be accepted for publication.
Author Response
Review Report (Reviewer 2)
Comments and Suggestions for Authors
It could be accepted for publication.
Authors thank ro the Reviewer 2 for the positive recommendation
